

# Temperature dependent rate coefficients for the reactions of the hydroxyl radical with the atmospheric biogenics isoprene, alpha-pinene and delta-3-carene

**Terry J. Dillon[1,2], Katrin Dulitz[1,3], Christoph B. M. Groß[1] and John N. Crowley[1]**

[1]Department of Air Chemistry, Max Planck Institute for Chemistry, 55124 Mainz, Germany

[2]now at Wolfson Atmospheric Chemistry Laboratories, Department of Chemistry, University of York, YO10 5DD, York, UK

[3]now at Physikalisches Institut, University of Freiburg, Hermann-Herder Str. 3, 79104 Freiburg, Germany

*Correspondence to*: Terry J. Dillon (terry.dillon@york.ac.uk)

**Abstract.** Pulsed laser methods for OH generation and detection were used to study atmospheric degradation reactions for three important biogenic gases: OH + isoprene (R1); OH + $\alpha$-pinene (R2); and OH + $\Delta$-3-carene (R3). Gas-phase rate coefficients were characterised by non-Arrhenius kinetics for all three reactions. For (R1), $k_1$(241-356 K) = (1.93 ± 0.08)×$10^{-11}$exp{(466 ± 12)/$T$} cm$^3$ molecule$^{-1}$ s$^{-1}$ was determined, with a room temperature value of $k_1$(297 K) = (9.3 ± 0.4)×$10^{-11}$ cm$^3$ molecule$^{-1}$ s$^{-1}$, independent of bath-gas pressure (5 – 200 Torr) and composition (M = N$_2$ or air). Accuracy and precision were enhanced by online optical monitoring of isoprene, with absolute concentrations obtained via an absorption cross-section, $\sigma_{isoprene}$ = (1.28 ± 0.06)×$10^{-17}$ cm$^2$ molecule$^{-1}$ at $\lambda$ = 184.95 nm, determined in this work. These results indicate that significant discrepancies between previous absolute and relative rate determinations of $k_1$ result in part from $\sigma$ values used to derive the isoprene concentration.

Similar methods were used to determine rate coefficients (in $10^{-11}$ cm$^3$ molecule$^{-1}$ s$^{-1}$) for (R2-R3): $k_2$(238 – 357 K) = (1.83 ± 0.04)×exp{(330 ± 6) / $T$}; and $k_3$(235 – 357 K) = (2.48 ± 0.14)×exp{(357 ± 17) / $T$}. This is the first temperature-dependent dataset for (R3) and enables the calculation of reliable atmospheric lifetimes with respect to OH removal for e.g. boreal forest springtime conditions. Room temperature values of $k_2$(296 K) = (5.4 ± 0.2)×$10^{-11}$ cm$^3$ molecule$^{-1}$ s$^{-1}$ and $k_3$(297 K) = (8.1 ± 0.3)×$10^{-11}$ cm$^3$ molecule$^{-1}$ s$^{-1}$ were independent of bath-gas pressure (7 – 200 Torr, N$_2$ or air), and in good agreement with previously reported values. In the course of this work, 184.95 nm absorption cross-sections were determined: $\sigma$ = (1.54 ± 0.08) × $10^{-17}$ cm$^2$ molecule$^{-1}$ for $\alpha$-pinene and (2.40 ± 0.12) × $10^{-17}$ cm$^2$ molecule$^{-1}$ for $\Delta$-3-carene.





## 1 Introduction

A large quantity and variety of volatile organic compounds (VOC) are released into the Earth's atmosphere from vegetation; such biogenic emissions exceed those of anthropogenic origin by a factor of ~ 10 (Goldstein and Galbally, 2007). Isoprene (2-methyl-1,3-butadiene, $C_5H_8$) contributes about 40 % (Guenther et al., 1995) of this gigatonne ($10^{15}$ g C) emission of

5 biogenic VOC. The hydroxyl radical, OH, is the primary oxidant in the troposphere, initiating the degradation of most important VOC such as $CH_4$, isoprene and the monoterpenes (two isoprene units, molecular formula = $C_{10}H_{16}$) (Atkinson and Arey, 2003). VOC lifetimes are thus constrained by OH, and its abundance is an important test of model reliability.

Much attention was recently focussed on chemistry over tropical forests, as these ecosystems influence the Earth's chemistry-climate system as the principal sink for the greenhouse gases $CO_2$ and $CH_4$ and source of isoprene and other VOC.

Models have traditionally predicted low levels of OH over tropical forests, due to its rapid reactions with VOC (notably isoprene, R1), and a lack of efficient OH regeneration mechanisms.

$$OH + isoprene \rightarrow (products) \qquad\qquad (R1)$$

Contrary to these predictions, results from field campaigns have demonstrated that a high OH concentration is maintained (Butler et al., 2008;Lelieveld et al., 2008;Martinez et al., 2010;Stone et al., 2011) the greatest discrepancies (factor of ≈ 3 to

15 10) between measured and modelled [OH] being observed where isoprene mixing ratios were large (Kubistin et al., 2010). The representation of photochemistry in models relies almost exclusively on lab-based measurements e.g. of absorption cross-sections, quantum yields, rate coefficients and product yields. For (R1), the rate coefficients $k_1(T)$ reported in the literature are characterized by small but significant differences, notably at temperatures away from 298 K. Accordingly, in the work presented in this paper, absolute rate coefficients $k_1(241 - 356$ K) were determined by highly-accurate methods at

20 realistic atmospheric temperature conditions.

The largest global contribution to atmospheric terpene emissions is from boreal forests covering Canada, Northern Europe and Siberia, together encompassing one third of the Earth's forests. Emissions peak in summer due to strong correlations with temperature and photosynthetic activity. However, similar emission maxima have been observed in the autumn and in late-winter / early spring when they are attributed to an increase in photorespiration rate (Back et al., 2005;Hakola et al.,

2003). Throughout the year shorter bursts of terpene emission, associated with damage induced by *eg*. animals (Heijari et al., 2011), pathogens, storms (Haase et al., 2011) and harvesting may impact on the local OH budget (Noelscher et al., 2012). Elsewhere, monoterpenes have been detected from other vegetation, notably the forest floor (Hellen et al., 2006) and over active phytoplankton blooms in the southern ocean (Yassaa et al., 2008).

Oxidation of terpenes proceeds to a vast number of gas- and condensed-phase products, with consequent production of

30 tropospheric ozone, peroxyacylnitrates and secondary organic aerosol (Atkinson and Arey, 2003). The product distribution from any given monoterpene depends upon physical conditions of temperature, pressure and humidity, together with the identity of the primary oxidant. Reactions with OH, $O_3$, and perhaps Cl initiate terpene degradation by day, with $NO_3$



contributing at night (Ng et al., 2017). The relative importance of these reactions depends on both the availability of the oxidant and the size of the corresponding rate coefficient $k(T)$. These rate coefficients are often reasonably well-established at around ambient temperature, though the database is sparse for terpene reactions for low temperature conditions ($T < 298$ K) most relevant to high, northern latitudes.

Summertime observations in the Finnish boreal forest (Williams et al., 2011) demonstrate that more than half the total terpene concentration measured above the boreal forest canopy was accounted for by two species: α-pinene (2,6,6-trimethylbicyclo-[3.1.1]hept-2-ene) and Δ-3-carene (3,7,7-trimethylbicyclo[4.1.0]hept-3-ene) (Hakola et al., 2012;Yassaa et al., 2008). Accordingly, this manuscript describes lab-based experiments to study the reactions (R2-R3) of OH with these two particularly important biogenic VOC:

OH + α-pinene  → (products)                                    (R2)

    OH + Δ-3-carene  → (products)                                    (R3)

The techniques employed here allowed the first absolute kinetic studies of (R2 & R3) at the temperatures most representative of boreal forests which can be as low as 253 K during late autumn and early spring months throughout which monoterpene emissions persist (Hakola et al., 2012).

## 2 Experimental

The technique of pulsed laser photolysis (PLP) radical generation coupled to pulsed laser induced fluorescence (LIF) detection of OH has been used in several absolute kinetic studies of OH + VOC reactions from this laboratory. Details of the apparatus and methods were published previously (Dillon et al., 2012;Karunanandan et al., 2007), and therefore only a brief

description is given here.

Experiments were conducted in a 500 cm$^3$ quartz reactor. Cell temperature was controlled by circulating a cryogenic fluid through an outer jacket, and monitored with a J-type thermocouple; pressure was monitored using capacitance manometers (MKS). Gas flow rates of up to 2000 cm$^3$ (STP) min$^{-1}$ ensured that a fresh gas sample was available for photolysis at each laser pulse. Fluorescence from OH was detected by a photomultiplier tube screened by a 309 nm interference filter and a BG

26 glass cut-off filter. Excitation of the $A^2\Sigma(v = 1) \leftarrow X^2\Pi(v = 0)$, $Q_{11}(1)$ transition at 282 nm was achieved using the frequency doubled emission from a Nd-YAG pumped dye laser. The detection limit at $P = 50$ mbar (air) was found to be ≈ $10^9$ molecule cm$^{-3}$ for a S/N = 1 (20 scans). An exciplex laser operating at 248 nm (KrF) was used to initiate radical chemistry via photolysis of $H_2O_2$ (R4).

    $H_2O_2 + h\nu$ (248 nm) → 2OH                                    (R4)





Typically a laser fluence of 12 mJ cm$^{-2}$ was used with [$H_2O_2$] = 1×10$^{14}$ molecules cm$^{-3}$, to generate [OH] ≈ 3×10$^{11}$ molecules cm$^{-3}$. Data analysis (see section 3) was considerably simplified by working under pseudo first-order conditions, of [$H_2O_2$] >> [OH], even when the concentrations of the organic reactant were low, OH then reacting mainly with $H_2O_2$.

$$OH + H_2O_2 \rightarrow HO_2 + H_2O \qquad\qquad (R5)$$

Time dependent OH profiles were obtained by recording the LIF signal intensity as the delay-time between the photolysis and the probe lasers was stepped from 0 to a maximum of 15 ms. Accuracy and precision of the rate coefficients determined in this work was further enhanced by online monitoring of the terpenoids isoprene, α-pinene and Δ-3-carene by VUV absorption spectroscopy. In order to achieve this, absorption cross-sections, $\sigma$, were determined experimentally by recording the attenuation of VUV light by static samples of each terpenoid. Two absorption cells were used, of optical path length $\ell$ =

30.4 cm (fitted with heated quartz windows) and $\ell$ = 1 cm. Concentrations were determined by manometric methods, with absolute cell-pressure measured to an accuracy of ± 0.01 Torr using a calibrated 10 Torr capacitance manometer. Samples were supplied either undiluted from the head-space above a liquid sample, or from a glass storage bulb as a dilute mixture (≈ 1 part in 1000 $N_2$). These two supply methods had complementary advantages and disadvantages. Use of undiluted samples was more direct, but with potentially large problems due to condensation onto windows, and accurate measurement of low

saturation vapour pressures. Such problems were minimized when using dilute VOC at higher pressure, but with the potential for errors associated with sample preparation and storage.

Collimated light from either a low-pressure Hg lamp ($\lambda$ = 184.95 nm) or a $D_2$ lamp (180 - 260 nm) transmitted the sample and was focussed onto the entrance slit of a 0.5 m monochromator equipped with a grating (300 lines mm$^{-1}$ blazed at 200 nm) and a diode-array detector. The volume around the lamp, cell, and monochromator was purged with $N_2$ to minimise

absorption by $O_2$, $O_3$ or $H_2O$. Spectra were recorded at an experimental resolution of 0.4 nm, estimated from full-width at half-maximum observed for either the $\lambda$ = 184.95 nm or 253.65 nm Hg-lines, as appropriate. Absorption cross-sections, $\sigma$ were obtained via the Beer-Lambert Law (Eq. 1).

$$I = I_0 \times \exp\{-\sigma\ \ell\ [\text{terpenoid}]\} \qquad\qquad \text{Eq. (1)}$$

where $I$ and $I_0$ are respectively the transmitted and incident light intensity and $\ell$ is the optical path length.

Online reagent monitoring in kinetic experiments was conducted downstream from the photolysis cell, using an absorption cell ($\ell$ = 43.8 cm) equipped with a low-pressure Hg lamp and a 185 nm interference filter. This setup was used to measure absolute reagent concentrations via VUV optical absorption at $\lambda$ = 184.95 nm. Following the application of small corrections for measured 254 nm stray light intensity (< 8%), and differences in $T$ (< 30 %) and $P$ (< 1%) between the cells, [terpenoids] of the order of 10$^{12}$ molecule cm$^{-3}$ could be measured (see section 3.1). Comparison of these optically derived concentrations

to approximate manometric measurements was used to confirm that no significant fraction of any of the terpenoids was deposited at the walls in low temperature experiments.

Chemicals: isoprene (Aldrich 99%, Sigma-Aldrich), α-pinene (Aldrich, 98%) and Δ-3-carene (Aldrich, 99%) were degassed by repeated freeze-pump-thaw cycles at 77 K, and diluted in $N_2$ for storage and supply. $H_2O_2$ (Roth, 35% by weight) was



concentrated by pumping away water and introduced to the experiment via a bubbler. $N_2$ and $O_2$ (Messer 99.999 %) were used without further purification.

## 3 Results and discussion

Detailed below are determinations of absorption cross-sections for the terpenoids isoprene, $\alpha$-pinene and $\Delta$-3-carene ($\sigma$, section 3.1), and temperature dependent rate coefficients for their reactions with OH: $k_1$ in section 3.2; and $k_2$ & $k_3$ in section 3.3. A discussion of possible systematic uncertainties, and a comparison of the results from this work with the existing literature is presented in section 3.4. All other uncertainties quoted throughout are precision only, representing two standard errors.

### 3.1 Absorption cross-sections ($\sigma$) for isoprene, $\alpha$-pinene and $\Delta$-3-carene

**Figure 1** displays (as the solid red line) the UV absorption spectrum of isoprene obtained in this work, together with results from two earlier studies. Cross sections were obtained at an experimental resolution of $\delta\lambda \approx 0.4$ nm (the full width half maximum of the $\lambda = 253.65$ nm Hg line) in the range of 194 - 226 nm and are reported in **Table 1** at 1 nm intervals. As is evident from **Figure 1**, there is excellent agreement ($\leq 1\%$ difference) between the results from this work and the more extensive and higher resolution ($\delta\lambda \approx 0.075$ nm) dataset from (Martins et al., 2009) depicted by the blue dotted line. (Campuzano-Jost et al., 2004) reported cross-sections (black dashed line on **Figure 1**) in the range 203 – 233 nm that are on average about 10% smaller. The reasons for this discrepancy are unclear, though it seems unlikely to have resulted from problems with sample purity which was > 99% in all three studies when supplied. Nor were differences in spectral resolution a likely source of error, since the highest-resolution study from (Campuzano-Jost et al., 2004) ($\delta\lambda \approx 0.05$ nm) reports the smallest differential cross-sections.

Also displayed on **Figure 1**, the red star depicts results from this work at $\lambda = 184.95$ nm that used a low-pressure Hg lamp as light source. **Figure 2** displays the corresponding plot of $\ln(I_0 / I) / \ell$ vs. [isoprene] which was used to obtain $\sigma_{\text{isoprene}}$ via (Eq. 1). Note the good agreement between data obtained across a wide range of concentrations, and over different optical path lengths. A proportional fit to all the data yields a value of $\sigma_{\text{isoprene}} = (1.28 \pm 0.02) \times 10^{-17}$ cm$^2$ molecule$^{-1}$ at $\lambda = 184.95$ nm, the wavelength used to derive [isoprene] in subsequent experiments (sections 3.2). Evident from **Figure 1** is the good agreement between this discrete value (the red star), and the literature spectrum from (Martins et al., 2009) who reported $\sigma_{\text{isoprene}} = 1.29 \times 10^{-17}$ cm$^2$ molecule$^{-1}$. Taking into account potential systematic errors from *eg*. manometric measurements and sample purity we estimate an overall uncertainty of $\approx 5$ %, *ie*. $\sigma_{\text{isoprene}} = (1.28 \pm 0.06) \times 10^{-17}$ cm$^2$ molecule$^{-1}$.

**Figure 3** displays similar plots of $\ln(I_0 / I) / \ell$ *vs*. [monoterpene] used to obtain (via Eq. 1) 184.95 nm absorption cross-sections for $\alpha$-pinene and $\Delta$-3-carene. A proportional fit to the $\alpha$-pinene data (blue-squares) yields a gradient $\sigma_{\alpha\text{-pinene}} = (1.54$





$\pm\ 0.05)\times10^{-17}$ cm$^2$ molecule$^{-1}$. Confidence in this result was enhanced by noting that over a wide range of concentrations, good agreement and proportionality was observed for data from both undiluted α-pinene samples, and those supplied dilute from a glass bulb (see section 2). Taking into account potential systematic errors we estimate an overall uncertainty of ≈ 5 %, *ie.* $\sigma_{\alpha\text{-pinene}}$ = $(1.54 \pm 0.08)\times10^{-17}$ cm$^2$ molecule$^{-1}$ at 184.95 nm. This value is significantly larger than that listed in the UV

spectrum (116 – 250 nm) of (Kubala et al., 2009) who report a value of $1.28\times10^{-17}$ cm$^2$ molecule$^{-1}$. The reasons for this ≈ 15% discrepancy are unclear. The Kubala spectrum does display some structure in the region 175 – 190 nm, though these features are orders of magnitude coarser than the reported experimental resolution (δλ = 0.075 nm), or the narrow-bandwidth (δλ < 0.001 nm) Hg-line light source used in this work. In the present study, cross-sections at λ = 184.95 nm were required to convert online optical measurements into absolute [α-pinene] for rate-coefficient determinations (see section 3.2). As the

light source used for α-pinene monitoring was also a narrow-bandwidth Hg-lamp, it is appropriate to use the $\sigma_{\alpha\text{-pinene}}$ value determined in this work.

Similar procedures were followed and a similar quality of data obtained (red circles in **Figure 3**) in the determination of $\sigma_{\Delta\text{-3-carene}}$ = $(2.40 \pm 0.03)\times10^{-17}$ cm$^2$ molecule$^{-1}$. There appear to be no previously published values for this parameter. It is significantly larger than $\sigma_{\alpha\text{-pinene}}$ and similar to values reported for other terpenes, eg. $\sigma$ (in $10^{-17}$ cm$^2$ molecule$^{-1}$) = 2.3 for β-

15 pinene (Kubala et al., 2009) and 4.6 for limonene (Smialek et al., 2012). Assessment of potential systematic errors results in an overall uncertainty of ≈ 5 %, *ie.* $\sigma_{\Delta\text{-3-carene}}$ = $(2.40 \pm 0.12)\times10^{-17}$ cm$^2$ molecule$^{-1}$ at λ = 184.95 nm.

### 3.2 Absolute determinations of $k_1$ (OH + isoprene)

PLP-LIF studies were carried under pseudo-first order conditions of [isoprene] >> [OH], such that the OH time profiles, $S(t)$, were described by a simple exponential decay, expression (Eq. 2):

$S(t) = S_0 \exp\{-Bt\}$                                               Eq. (2)

The parameter $S_0$ describes (in arbitrary units) the LIF signal at $t = 0$ and is proportional to the initial [OH] produced by the laser pulse (R4). The parameter $B$ is the pseudo first-order rate coefficient for OH decay (and includes components from both reactive and transport losses). **Figure 4** displays typical OH decay profiles recorded in the presence of three different excess [isoprene], whilst other conditions of $P$ = 100 mbar (N$_2$), $T$ = 298 K and [H$_2$O$_2$] ≈ $1\times10^{14}$ molecule cm$^{-3}$ were unchanged.

The OH decays were exponential over at least two orders of magnitude and were fit with Eq. (2) to yield values of $B$ with a high degree of precision (errors generally less than ± 2%). Systematic errors from unwanted radical chemistry were unlikely, since isoprene absorbs only weakly at the photolysis wavelength ($\sigma$ = $1.18\times10^{-19}$ cm$^2$ molecule$^{-1}$ at 248 nm from Martins et al.), whilst the use of low [OH] ensured that losses of OH by reaction with itself or with products from (R1) or (R5) could be disregarded in this analysis. Nonetheless, test experiments were conducted using an invariant [isoprene] = $3.2\times10^{14}$ molecule

30 cm$^{-3}$ whilst photolysis laser fluence, and thus radical concentrations were varied by a factor of fifteen. No systematic





influence was discerned (< 1% scatter in $B$ parameters) from such test experiments conducted both in $N_2$ and in air, reinforcing confidence in these results.

**Figure 5** displays plots of $B$ vs. [isoprene] at three temperatures which were fit (Eq. 3) to obtain the bimolecular rate coefficients $k_1(T)$.

$\qquad$ $B = k_1[\text{isoprene}] + k_{\text{loss}}$ $\qquad\qquad\qquad\qquad\qquad\qquad\qquad$ Eq. (3)

where $k_1$ is the bimolecular rate coefficient of interest. The term $k_{\text{loss}}$ ($s^{-1}$) accounts for other losses of OH, here dominated by reaction (R5) with the photolysis precursor $H_2O_2$, with some small contribution from diffusion and flow out of the reaction zone.

The data display good linearity, with the slopes identified as $k_1(T)$ at three different temperatures, and intercept values (around $B = 200$ $s^{-1}$) in line with the predicted loss rate for reaction with $H_2O_2$ (R5). A mean of six values obtained at room temperature yields $k_1(297\ \text{K}) = (9.3 \pm 0.2) \times 10^{-11}$ $cm^3$ molecule$^{-1}s^{-1}$. Within the experimental uncertainty, the measured rate coefficients were found to be independent of the bath gas pressure ($3.8 < P$ / Torr $< 165$) or identity ($N_2$ or air). Experimental conditions and results of all such determinations of $k_1$ are listed in **Table 2**. Note that the optical derivation of [isoprene] at 184.95 nm means that ratios of $k_1$ / $\sigma_{\text{isoprene}}$ were actually determined, and that any subsequent improvements in the database for isoprene cross-sections may be used to update the kinetic results from this work. Taking into account potential systematic errors, principally from $\sigma_{\text{isoprene}}$, and a small (2 to 4 %) uncertainty in carrying out the optical measurements we estimate a realistic overall error of $\pm 5\%$, or $k_1(297\ \text{K}) = (9.3 \pm 0.4) \times 10^{-11}$ $cm^3$ molecule$^{-1}s^{-1}$.

The data displayed in **Figure 5** further reveal the "negative" or non-Arrhenius temperature dependence for $k_1$, with the largest slopes obtained in experiments conducted at the lowest temperatures (241 K). This trend for $k_1(T)$ is well-established in the literature; results from these previous studies, together the $k_1$ values from this work (listed in **Table 2**) are depicted on **Figure 6**.

### 3.2.1 Comparison of $k_1$ with literature values

The kinetic database for (R1) displays some inconsistencies, with results for $k_1(298\ \text{K})$ from a recent absolute determination by (Campuzano-Jost et al., 2004) some 15% smaller than other values reported, mostly from relative rate studies (Atkinson et al., 2006b). The ambient temperature results from this work lie in between these two extremes. Absolute determinations of $k_1$ were generally conducted at lower pressure than the majority of the relative rate studies, and could therefore yield values that underestimate $k_1(1\ \text{bar}, 298\ \text{K})$ if the reaction was not at the high-pressure limit under all experimental conditions. However, results from this work revealed no systematic change in $k_1$ over the range pressure from 3.8 to 165 Torr, in good qualitative agreement with the results from (Campuzano-Jost et al., 2004) (50 to 600 Torr, $N_2$ or air). Two studies of $k_1(298\ \text{K})$ conducted at lower pressures indicate that the high-pressure limit was close to 3 Torr (He) (Singh and Li, 2007) and 1 Torr (He) (Chuong and Stevens, 2000). In contrast, North and coworkers observed fall-off behaviour for $k_1(298\ \text{K})$ over the pressure range $0.5 < P(\text{Ar})$ / Torr $< 20$, in line with their CVTST / RRKM calculations (McGivern et al., 2000;Park et al.,





2004). Whilst no low-pressure studies of $k_1(P)$ using $N_2$ or air bath gas are available, the consensus would appear to be that the $k_1$(298 K) values reported in this work at up to $P$ = 165 Torr ($N_2$) and by (Campuzano-Jost et al., 2004) are close to the high pressure limit, representative of $k_1$(760 Torr) and thus directly comparable to the relative rate results in the literature.

The experimental methods used in this work were similar to those employed by (Campuzano-Jost et al., 2004) who obtained

$k_1$(297 K) = (8.47 ± 0.59)×$10^{-11}$ cm$^3$ molecule$^{-1}$s$^{-1}$, lower than most other determinations, and outside the range of the IUPAC recommendation. (Campuzano-Jost et al., 2004) used a cross section of 8.29×$10^{-18}$ cm$^2$ molecule$^{-1}$ at 228.8 nm when determining [isoprene] from optical measurements. The extensive high-resolution dataset from Martins *et al.* provides cross sections at both 228.8 nm and at 184.95 nm (used in this work). If cross-sections from Martins et al. are used to derive [isoprene] for the (Campuzano-Jost et al., 2004) results, a value of $k_1$(297 K) = (9.2 ± 0.7)×$10^{-11}$ cm$^3$ molecule$^{-1}$ s$^{-1}$ is

obtained (see also the filled blue circle datapoints on **Figure 6**), consistent with results from this work. A similar re-evaluation using cross-sections from Martins et al. does not change the rate coefficients from our study.

Evaluation of all ambient temperature literature to date led IUPAC to recommend a room temperature rate coefficient of $k_1$(298 K) = 1.0×$10^{-10}$ cm$^3$ molecule$^{-1}$s$^{-1}$, slightly larger than the upper limit to our determination, though the two values overlap given IUPAC quotes an uncertainty of $\Delta\log(k_1)$ = 0.06, ie. (in $10^{-11}$ cm$^3$ molecule$^{-1}$ s$^{-1}$): 8.71 < $k_1$(298 K) < 10.15.

The results obtained in this work, together with the re-calculation of the results from (Campuzano-Jost et al., 2004) would indicate that this evaluation is both a slight overestimate of the true value of $k_1$(298 K), and is quoted with an overly conservative uncertainty.

The results from previous studies of $k_1(T)$ together with the IUPAC recommendation and the data from this work (red squares) are presented in Arrhenius format on **Figure 6**. All temperature dependent investigations characterized $k_1(T)$ with a

non-Arrhenius temperature dependence, in qualitative agreement with results from this work. However, there are surprisingly few published studies of (R1) over a range of temperatures characteristic of the troposphere, with only (Campuzano-Jost et al., 2004) and (Park et al., 2004) reporting values of $k_1(T < 298$ K). As is evident from **Figure 6**, the spread of $k_1(T < 298$ K) is large, around ± 30% from the uncorrected values from (Campuzano-Jost et al., 2004) to the results from (Park et al., 2004) at $T$ = 279 K. IUPAC have completed an extensive evaluation of this literature, together with

unpublished $k_1$(249 – 438 K) results from Siese and co-workers and many published values at $T \approx$ 298 K (not displayed in Figure 8 for reasons of clarity). The Arrhenius expression (Eq. 4) was used to parameterise $k_1$

$$k_1(T) = A \times \exp\{ B / T \} \hspace{3cm} \text{Eq. (4)}$$

with recommended parameters of $k_1$(240-430 K) = 2.7×$10^{-11}$exp{390/$T$} cm$^3$ molecule$^{-1}$ s$^{-1}$, with associated uncertainties of $\Delta\log k_1$ of ± 0.06 at 298 K and $\Delta B$ of ± 100 K to account for the scarcity and spread of $k_1(T)$ data. A weighted fit of

expression (E4) to the data from this work yields $k_1$(241-356 K) = (1.93 ± 0.08)×$10^{-11}$exp{(466 ± 12)/$T$} cm$^3$ molecule$^{-1}$ s$^{-1}$, an adequate parameterisation over this limited range of temperature. The good agreement between the experimental results obtained in this work, the corrected values from (Campuzano-Jost et al., 2004) and the IUPAC recommendation would indicate that the recommended uncertainties should be reduced in any subsequent re-evaluation by IUPAC. Results from



(Spangenberg et al., 2004), with $k_1$(58 – 300 K) reaching a maximum value of ~ $3\times10^{-10}$ cm$^3$ molecule$^{-1}$ s$^{-1}$ at around $T$ = 120 K, demonstrate that large errors could result if the Arrhenius expressions presented here and by IUPAC were extrapolated beyond the temperature range stated.

### 3.3 Absolute determinations of $k_2$ & $k_3$

**Figure 7** displays OH LIF profiles recorded in the presence of three different excess α-pinene concentrations, with conditions of $P$ = 97 Torr (air), $T$ = 296 K and [H$_2$O$_2$] ≈ $1\times10^{14}$ molecule cm$^{-3}$ otherwise unchanged. The OH decays were exponential over approximately two orders of magnitude and were therefore fit with expression (E2) to yield $B$ parameters with a high degree of precision (± 5% or less). As with the determinations of $k_1$ detailed above, systematic errors due to secondary chemistry were considered unlikely, since monoterpenes absorb only weakly at the photolysis wavelength (the

248 nm cross section is < $1\times10^{-20}$ cm$^2$ molecule$^{-1}$ (Kubala et al., 2009), resulting in photolysis of < $1\times10^{11}$ molecule cm$^{-3}$) of α-pinene. Furthermore, use of small OH concentrations and hence low conversion of α-pinene ensured that perturbations to the LIF profiles by reaction of OH with itself or with the products of (R2) or (R5) could be disregarded in this analysis. All data for R2 and R3 recorded in this work were of similar or superior quality to that presented in **Figure 7**, where the presence of the efficient fluorescence quencher O$_2$ and relatively large bath gas pressures, had a negative impact on LIF

signal quality.

The red squares in **Figure 8** represent results from a series of experiments conducted at $P$ = 97 Torr (air) and $T$ = 296 K. Values of $B$ were plotted against [α-pinene] calculated from its optical extinction at 184.95 nm (see section 2). A weighted linear fit (using statistical errors in $B$) yields a gradient of $k_2$(296 K) = (5.3 ± 0.2)$\times10^{-11}$ cm$^3$ molecule$^{-1}$s$^{-1}$. The experimental conditions used and results obtained for all determinations of $k_2$ are listed in **Table 1**. A mean of the four values obtained at

around room temperature yields $k_2$(296 K) = (5.4 ± 0.2)$\times10^{-11}$ cm$^3$ molecule$^{-1}$s$^{-1}$. Within experimental uncertainty, these rate coefficients were independent of the bath gas pressure (7 < $P$ / Torr < 197) or identity (N$_2$ or air). This result is in reasonable agreement with a number of experimental and theoretical determinations of $k_1$ at around ambient temperature, summarized in **Table 3**.

Experiments conducted at various temperatures revealed a "negative" or non-Arrhenius temperature dependence for $k_2$, with

the largest values obtained in experiments conducted at the lowest temperatures. This trend for $k_2(T)$ is well-established by four literature datasets (Chuong and Stevens, 2000;Gill and Hites, 2002;Kleindienst et al., 1982;Montenegro et al., 2012), and is commonly observed for other radical additions to alkenes. Results from these previous studies, together with the $k_2$ values from this work (listed in **Table 2**) are depicted on **Figure 9**.

A similar methodology was followed to determine $k_3(T)$ for the reaction of OH + Δ-3-carene. LIF profiles (not shown) were

of a similar quality to those depicted on **Figure 7**, and were fit (E2) to obtain $B$ parameters. Optical signals were converted (E1) to absolute [Δ-3-carene] using ℓ = 43.8 cm and a value of $\sigma_{\text{Δ-3-carene}}$ = $2.4\times10^{-17}$ cm$^2$ molecule$^{-1}$ determined in this work (section 3.1). **Figure 8** displays (blue circles) a plot of $B$ vs. [Δ-3-carene] for a series of experiments conducted at $T$ = 296 K




and $P$ = 96 Torr (air). The rate coefficient for (R3) is clearly larger than that determined for (R2) above; a weighted linear fit yields a gradient $k_3$(296 K) = (8.1 ± 0.1)×10$^{-11}$ cm$^3$ molecule$^{-1}$ s$^{-1}$. Experiments were repeated using different bath gas pressures and in N$_2$ (see **Table 1**). No trends in $k_3$ were observed and a mean of these five results yields $k_3$(297 ± 1) = (8.1 ± 0.2)×10$^{-11}$ cm$^3$ molecule$^{-1}$ s$^{-1}$. Results from all experiments, depicted as black circles on **Figure 9**, demonstrate for the first

time the non-Arrhenius temperature dependence for $k_3(T)$.

### 3.3.1 Comparison of $k_2$ and $k_3$ with literature values

**Table 2** presents a comparison of the ambient temperature values of $k_2$ and $k_3$ determined in this work with the available literature. Regarding $k_2$, there is excellent agreement between the results of this work and those obtained by the relative rate technique (Atkinson et al., 1986;Gill and Hites, 2002;Montenegro et al., 2012;Winer et al., 1976). The principal source of

error in relative rate determinations often derives from the choice of reference rate coefficient, required to convert an experimentally derived rate coefficient ratio into an absolute rate coefficient. Confidence may be derived from the fact that five different reference reactions were used across the four published relative rate (R2) studies. Absolute rate coefficient determinations do not depend upon literature rates, but do require an accurate knowledge of concentrations of the excess reagent (α-pinene or Δ-3-carene). The present dataset used in-situ optical absorption spectroscopy to accurately measure [α-

pinene], and subsequently calculate an absolute $k_2$(296 K) = (5.4 ± 0.2)×10$^{-11}$ cm$^3$ molecule$^{-1}$ s$^{-1}$. Any error in $\sigma_{\alpha-pinene}$ propagates directly into $k_2$. Taking into account potential systematic errors, principally the 5% uncertainty in $\sigma_{\alpha-pinene}$ (section 3.1), a similar uncertainty in carrying out the optical measurements, and 2% for sample purity, a total uncertainty of around 8% was calculated so we quote: $k_2$(296 K) = (5.4 ± 0.5)×10$^{-11}$ cm$^3$ molecule$^{-1}$ s$^{-1}$. Within combined uncertainties there is reasonable agreement between this result and three previous absolute kinetic studies of (R2) (Chuong et al., 2002;Davis and

Stevens, 2005;Kleindienst et al., 1982), all of which relied on manometric determinations of [α-pinene] to derive $k_2$. The IUPAC data evaluation panel currently recommends a value of $k_2$(298 K) = 5.3×10$^{-11}$ cm$^3$ molecule$^{-1}$ s$^{-1}$ (Ammann et al., 2016;Atkinson et al., 2006b), based on results from (Atkinson et al., 1986;Gill and Hites, 2002;Kleindienst et al., 1982), with an associated Δlog$k_2$ = 0.15 or approximately +40%, -30%. A re-evaluation taking into account the data from this work and from the recent (Montenegro et al., 2012) study yields (via a weighted mean) $k_2$(298 K) = 5.4×10$^{-11}$ cm$^3$ molecule$^{-1}$ s$^{-1}$; a

considerably smaller Δlog($k_2$) of 0.06 (around ± 15%) would encompass all experimentally determined ambient temperature $k_2$ values.

IUPAC (Ammann et al., 2016;Atkinson et al., 2006b) use the Arrhenius expression (E4) to parameterise $k_2(T)$. A weighted fit of expression (E4) to the data from this work yields $k_2$(238-357 K) = (1.83 ± 0.04)×10$^{-11}$exp{(330 ± 6)/$T$} cm$^3$ molecule$^{-1}$ s$^{-1}$, represented by the black dot-dash line on **Figure 9**. Where data overlap (290 K to 357 K), these $k_2(T)$ lie within the large

uncertainty (Δlog$k_2$ = 0.15 at 298 K; Δ$B$ = 200 K) reported by IUPAC for $k_2$(290 – 430 K) = 1.2×10$^{-11}$exp{440/$T$} cm$^3$ molecule$^{-1}$ s$^{-1}$ (Ammann et al., 2016;Atkinson et al., 2006b). Represented on **Figure 9** by the solid orange line, this recommendation was based on the results of (Gill and Hites, 2002;Kleindienst et al., 1982), and is not in good agreement





with the considerably smaller $k_2(T > 350$ K) from (Chuong et al., 2002). The reasons for this discrepancy are unclear. Chuong suggested that for their experimental conditions of $P = 5$ Torr (He), observed $k(T > 350$ K) may not be high-pressure limited values. However, no evidence for such a pressure-dependence in $k_2$ was observed in subsequent relative rate determinations of $k_2(T = 340$ K, $P = 1 - 5$ Torr He) from (Montenegro et al., 2012). This discrepancy is not resolved by any

results from the present study, as the apparatus did not permit experiments at sufficiently low pressures. The results presented in **Table 2** and elsewhere would indicate that the values of $k_2$ obtained here at $P > 7$ Torr and $T \approx 298$ K were in the high-pressure limit.

The current IUPAC recommendation (Ammann et al., 2016;Atkinson et al., 2006b) does not extend to important atmospheric temperatures lower than 290 K, as no data were available when the evaluation was made. However, there are now two

complementary datasets available. The absolute values from this work (see **Table 2**), and results from the most recent literature dataset (Montenegro et al., 2012) derived by relative-rate methods are in excellent agreement for $T < 298$ K conditions. **Figure 9** displays (the dot-dashed black line) results from a weighted (E4) fit to these two datasets: $k_2(238 - 298$ K) $= 1.51 \times 10^{-11}$ exp$\{378 / T\}$ cm$^3$ molecule$^{-1}$ s$^{-1}$. This formulation, which could be used to recommend $k_2(T)$ values for low temperatures was not extended to $T > 298$ K where significant uncertainties and literature discrepancies do remain.

A similar analysis was applied to $k_3$, whereby potential systematic errors (5% each for $\sigma_3$ and the optical measurements, 1% for sample purity) were combined with the statistical uncertainties listed in **Table 2** to yield an overall error estimate of around 8%, hence $k_3(298$ K) $= (8.2 \pm 0.6) \times 10^{-11}$ cm$^3$ molecule$^{-1}$ s$^{-1}$. The only previous determination of $k_3$ was from (Atkinson et al., 1986), who used the relative rate method to obtain $k_3(294$ K) $= (8.7 \pm 0.4) \times 10^{-11}$ cm$^3$ molecule$^{-1}$ s$^{-1}$. The good agreement between these two datasets, derived by very different experimental methods, is encouraging. It is clear from

the data presented in **Table 2** and displayed graphically on **Figure 9** that (R3) is characterized by negative or non-Arrhenius $k_3(T)$. A weighted (E4) fit yields $k_3(235 - 357$ K) $= (2.48 \pm 0.14) \times$exp$\{(357 \pm 17) / T\}$, depicted on **Figure 9** as the dashed black line. These results are the first reported for $k_3$ that were not obtained at room temperature. The non-Arrhenius behaviour observed is common for OH + monoterpene reactions that proceed predominantly by addition of OH to the unsaturated C=C groups, with a smaller contribution ($\sim$ 10%) from direct abstraction of allyl-stabilized hydrogens (Peeters et

al., 2001;Vereecken et al., 2007). To a first approximation therefore, (R2) and (R3) should proceed at quite similar rates, as the main reactive site (a methyl substituted endo-cyclic C=C group) is common to $\alpha$-pinene and $\Delta$-3-carene. The data listed in **Table 2** show that (R2) and (R3) are characterized by rate coefficients broadly consistent with this picture, ie. non-Arrhenius $k(T)$ values approaching the collision limit. Note that as discussed above, in the reaction conditions employed here, addition of OH to such large (C10) molecules need not be characterized by any discernable pressure-dependence. It is

somewhat surprising however that the results from this work and from (Atkinson et al., 1986) clearly demonstrate that OH reacts some 50% faster with $\Delta$-3-carene than with $\alpha$-pinene. This is unlikely to be due to any enhancement in abstraction rates as $\Delta$-3-carene has fewer allyl hydrogens available than does $\alpha$-pinene. The observed change in reactivity is presumably due to influence on the C=C reactive site by the different location of the $(CH_3)_2C$ substituent in $\alpha$-pinene and $\Delta$-3-carene.





Note that systematic attempts to parameterize $k_2$ *via* structure activity relationships have also failed at this level of detail (see *eg*. the predicted $k_2$ and $k_3$ in **Table 3** from (Vereecken et al., 2007) and from (Grosjean and Williams, 1992).

Finally, it should be reiterated that all $k_2$ and $k_3$ values reported in this work were determined *via* optically derived [monoterpene], and may more formally be considered as ratios such as $k_2(T) / \sigma_{\alpha\text{-pinene}}$ and $k_3(T) / \sigma_{\Delta\text{-3-carene}}$. Currently the database for 184.95 nm cross sections is sparse, consisting of only the results from this work and one other reported value for $\sigma_{\alpha\text{-pinene}}$ from (Kubala et al., 2009). Any subsequent improvements in the database for 184.95 nm absorption cross-sections for monoterpenes should be used to obtain up to date rate coefficients $k_2(T)$ and $k_3(T)$.

**4 Atmospheric implications and conclusions**

Breakdown of isoprene, initiated by reaction with the hydroxyl radical is one of the most important oxidation reactions in the troposphere. The spectroscopic and kinetic results obtained in this work indicate that significant discrepancies between previous absolute and relative determinations of $k_1$ were most likely the result of using erroneous absorption cross sections to derive [isoprene] in absolute studies. The value of $k_1(297\ \text{K}) = (9.3 \pm 0.4)\times10^{-11}\ \text{cm}^3\ \text{molecule}^{-1}\ \text{s}^{-1}$ reported here is in satisfactory agreement with many previous determinations using different methods, and confirms an atmospheric lifetime for isoprene of a few hours in typical rainforest conditions.

Absolute rate coefficients were determined for the gas-phase reactions of OH with two monoterpenes of atmospheric interest: $\alpha$-pinene (R2) and $\Delta$-3-carene (R3). At around ambient temperature, $k_2$ and $k_3$ were large and independent of bath-gas pressure (7 – 200 Torr) or composition ($N_2$ or air), indicating that these predominantly addition reactions are approaching their high-pressure limiting rate. The low temperature rate coefficients derived in this work may be used to calculate atmospheric lifetimes for monoterpenes in realistic boreal forest conditions. (Montenegro et al., 2012) used $T = 277\ \text{K}$ and $[\text{OH}] = 8.1\times10^5\ \text{molecule cm}^{-3}$ to calculate $\tau_2 = \{k_2(T) \times [\text{OH}]\}^{-1} = 5.8$ hours. The values of $T$ and [OH] used would appear appropriate for daytime boreal forest conditions during a springtime emission event (Back et al., 2005). When using the absolute value of $k_2(277\ \text{K})$ from this work, the calculated value of $\tau_2 = 5.8$ hours is unchanged, a result that significantly enhances confidence in this parameter. We note that similar calculations (using mixing ratios of 30 ppb for $O_3$ and kinetic data from IUPAC) demonstrate that $\alpha$-pinene is oxidised by $O_3$ at a very similar rate. The publication in this work of the first $k_3(T)$ allows the calculation of realistic atmospheric lifetimes for $\Delta$-3-carene with respect to (R3). Using $[\text{OH}] = 8.1\times10^5\ \text{molecule cm}^{-3}$ and $k_3(277\ \text{K}) = 9.0\times10^{-11}\ \text{cm}^3\ \text{molecule}^{-1}\ \text{s}^{-1}$ from this work, $\tau_3 = 3.8$ hours was calculated. The corresponding calculation (again using 30 ppb of $O_3$ and kinetic data from IUPAC) results in a loss rate some 2.5 slower for $\Delta$-3-carene with $O_3$. These results demonstrate that the dominant daytime loss process for $\Delta$-3-carene is reaction with OH (R3).





**Author contribution**

JNC & TJD designed all experiments and planned the overall project; KD conducted the spectroscopic experiments; TJD & CBMG conducted the kinetic experiments; TJD prepared the manuscript with contributions from all authors.

**Acknowledgements**

The authors thank Gerhard Schuster for ever-excellent technical support, Jonathan Williams & Luc Vereecken for helpful discussions regarding the monoterpene reactions, and the Max Planck Society for support *via* a research fellowship.

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



**Table 1: UV absorption cross-sections for isoprene from this work**

| $\lambda$ / nm [a] | $\sigma$ [b] | $\lambda$ / nm | $\sigma$ [b] | $\lambda$ / nm | $\sigma$ [b] | $\lambda$ / nm | $\sigma$ [b] |
|---|---|---|---|---|---|---|---|
| 184.95 [*] | 1.28 | 204 | 4.72 | 215 | 7.73 | 226 | 2.75 |
| 194 | 1.66 | 205 | 5.13 | 216 | 7.88 | 227 | 1.92 |
| 195 | 1.85 | 206 | 5.50 | 217 | 7.44 | 228 | 1.31 |
| 196 | 2.07 | 207 | 5.83 | 218 | 6.62 | 229 | 0.90 |
| 197 | 2.33 | 208 | 6.38 | 219 | 5.86 | 230 | 0.62 |
| 198 | 2.62 | 209 | 6.84 | 220 | 5.46 | 231 | 0.44 |
| 199 | 3.04 | 210 | 6.93 | 221 | 5.43 | 232 | 0.32 |
| 200 | 3.29 | 211 | 6.81 | 222 | 5.59 | 233 | 0.24 |
| 201 | 3.58 | 212 | 6.69 | 223 | 5.50 | 234 | 0.19 |
| 202 | 3.99 | 213 | 6.80 | 224 | 4.84 | | |
| 203 | 4.36 | 214 | 7.23 | 225 | 3.78 | | |

Notes: [a] data obtained at a resolution of $\Delta\lambda \approx 0.4$ nm, averaged and reported at 1 nm intervals except for [*] $\sigma$ at 184.95nm from high-resolution experiment (see section 3.1 & Fig. 2); [b] $\sigma$ in units of $10^{-17}$ cm$^2$ molecule$^{-1}$.



**Table 2: Summary of rate coefficients $k_1$, $k_2$ & $k_3$ determined in this work**

| $T$ / K | $P$ / Torr [a] | [terpene] [b] | $n$ [c] | $k$ / $10^{-11}$ [d] |
|---|---|---|---|---|
| **OH + isoprene → (products)    (R1)** | | | | |
| 241 | 6.0 | 22 – 222 | 5 | 12.9 ± 0.2 |
| 241 | 6.1 * | 22 – 222 | 5 | 13.2 ± 0.3 |
| 268 | 6.7 | 21 – 187 | 6 | 11.3 ± 0.14 |
| 298 | 3.8 | 39 – 287 | 5 | 9.20 ± 0.06 |
| 297 | 7.5 | 27 – 200 | 6 | 9.43 ± 0.14 |
| 298 | 42.9 | 72 – 438 | 8 | 9.19 ± 0.05 |
| 298 | 71.0 | 29 – 223 | 7 | 9.12 ± 0.10 |
| 298 | 103 | 57 - 574 | 7 | 9.20 ± 0.05 |
| 298 | 103 * | 112 - 574 | 5 | 9.22 ± 0.09 |
| 296 | 165 | 19 – 168 | 7 | 9.1 ± 0.4 |
| 296 | 165 * | 19 – 168 | 7 | 9.2 ± 0.4 |
| 330 | 8.1 | 15 – 236 | 6 | 7.91 ± 0.12 |
| 356 | 9.0 | 25 – 216 | 6 | 6.92 ± 0.13 |
| **OH + α-pinene → (products)    (R2)** | | | | |
| 238 | 6.0 | 7 - 90 | 7 | 7.33 ± 0.04 |
| 241 | 211 | 47 - 357 | 6 | 7.5 ± 0.2 |
| 246 | 96 * | 131 - 554 | 4 | 7.23 ± 0.10 |
| 246 | 97 | 131 - 554 | 4 | 6.94 ± 0.05 |
| 256 | 6.5 | 18 - 84 | 6 | 6.76 ± 0.07 |
| 274 | 7.0 | 8 - 30 | 5 | 6.12 ± 0.14 |
| 296 | 7.5 | 5 - 65 | 7 | 5.50 ± 0.08 |
| 296 | 96 | 43 - 464 | 5 | 5.38 ± 0.06 |
| 296 | 97 * | 43 - 464 | 5 | 5.25 ± 0.09 |
| 296 | 197 | 38 - 420 | 6 | 5.39 ± 0.04 |
| 323 | 7.2 | 19 - 157 | 7 | 5.44 ± 0.08 |
| 357 | 6.9 | 24 - 240 | 7 | 4.82 ± 0.05 |
| **OH + Δ-3-carene → (products)    (R3)** | | | | |
| 235 | 7.5 | 4 - 43 | 6 | 11.9 ± 0.3 |




| | | | | |
|---|---|---|---|---|
| 241 | 211 | 13 - 99 | 6 | 11.4 ± 0.3 |
| 271 | 96 | 44 - 530 | 5 | 9.11 ± 0.13 |
| 296 | 95 | 42 - 377 | 8 | 8.01 ± 0.13 |
| 296 | 95 * | 42 - 377 | 8 | 8.08 ± 0.13 |
| 298 | 7.4 | 8 - 63 | 7 | 8.10 ± 0.10 |
| 298 | 48 | 9 - 90 | 8 | 8.27 ± 0.08 |
| 298 | 205 | 34 - 262 | 9 | 8.1 ± 0.2 |
| 323 | 7.1 | 8 - 66 | 6 | 7.9 ± 0.2 |
| 357 | 6.9 | 10 - 99 | 7 | 6.92 ± 0.12 |

Notes: [a] bath gas identity M = $N_2$ unless marked [*] where M = air; [b] units of [terpene] are $10^{12}$ molecules $cm^{-3}$, with values derived from optical measurements using 185 nm σ values determined in this work; [c] $n$ = number of different [terpene] used in determination of $k$; [d] units of $k$ are $10^{-11}$ $cm^3$ $molecule^{-1}$ $s^{-1}$, uncertainties quoted are statistical only (two standard errors) derived from weighted linear fits (see Figs. 5 & 8).



**Table 3: Comparison with literature determinations of $k_2$ and $k_3$ at around 298 K**

| $k$ [a] | $T$ / K | Reference | Techniques used [b] |
|---|---|---|---|
| **Absolute determinations of $k_2$ (OH + α-pinene)** | | | |
| 5.4 ± 0.2 [c] | 296 | This work | PLP-LIF |
| 6.09 ± 0.3 | 300 | (Davis and Stevens, 2005) | DF-LIF |
| 6.08 ± 0.4 | 300 | (Chuong et al., 2002) | DF-LIF |
| 6.01 ± 0.82 | 298 | (Kleindienst et al., 1982) | FP-RF |
| **Relative rate determinations of $k_2$** | | | |
| 5.33 ± 0.79 | 298 | (Dash et al., 2014) | GC, *vs*, propene |
| 5.38 ± 0.3 | 298 | (Renbaurn-Wolff and Smith, 2012) | MS, *vs*. isoprene |
| 5.55 ± 0.97 | 298 | (Montenegro et al., 2012) | DF-MS, *vs*. isoprene |
| 5.0 ± 1.1 | 298 | (Gill and Hites, 2002) | MS, *vs*. 1-butene, 2-methylpropene & trans-2-butene |
| 5.35 ± 0.31 | 294 ± 1 | (Atkinson et al., 1986) | GC, *vs*. 2,3-dimethyl-2-butene |
| 5.8 ± 1.7 [d] | 305 | (Winer et al., 1976) | GC, *vs*. 2-methylpropene |
| **Rate theory for $k_2$** | | | |
| 4.65 | 298 | (Dash et al., 2014) | CVTST |
| 5.1 | 298 | (Vereecken et al., 2007) | Estimate using SAR |
| 5.1 | 300 | (Fan et al., 2005) | CVTST |
| 12 | 298 | (Grosjean and Williams, 1992) | Estimate using SAR |
| **Absolute determinations of $k_3$ (OH + Δ-3-carene)** | | | |
| 8.1 ± 0.3 [c] | 297 ± 1 | This work | PLP-LIF |
| **Relative Rate Determinations of $k_3$** | | | |
| 8.7 ± 0.4 | 294 ± 1 | (Atkinson et al., 1986) | GC, *vs*. 2,3-dimethyl-2-butene |
| **Rate theory for $k_3$** | | | |
| 0.506 | 298 | (Baptista et al., 2014) | QM calculations |
| 8.51 | 298 | (Peeters et al., 2007) | Estimate using SAR |
| 9.1 | 298 | (Grosjean and Williams, 1992) | Estimate using SAR |

Notes: [a] units of $k_1$ are $10^{-11}$ cm$^3$ molecule$^{-1}$ s$^{-1}$; [b] key to abbreviations PLP = pulsed laser photolysis, LIF = laser induced

fluorescence, FP = conventional flash photolysis, RF = resonance fluorescence, DF = discharge flow, GC = gas

chromatography, MS = mass-spectrometry, CVTST = canonical variational transition state theory, SAR = structure activity





relationship; [c] calculated using $\sigma$ values from this work. Realistic uncertainty estimate is $\pm$ 8 % for $k_1$ & $k_2$ (includes systematic errors in $\sigma$ and kinetic procedure, see section 3.3); [d] = recalculated using latest $k$(298 K) = $5.1\times10^{-11}$ cm$^3$ molecule$^{-1}$ s$^{-1}$ from IUPAC (Ammann et al., 2016;Atkinson et al., 2006a) for reference reaction OH + iso-butene.

**Figure 1**: UV absorption spectra for isoprene from this work and from the literature. The red solid line represents data recorded at a resolution of $\pm$ 0.4 nm; the red star depicts $\sigma_{\text{isoprene}}$ = (1.28 $\pm$ 0.02)$\times10^{-17}$ cm$^2$ molecule$^{-1}$ at 184.95 nm (see **Figure 2**) subsequently used to derive absolute concentrations for kinetic experiments. Results from this work agree well with the more extensive, high-resolution dataset from (Martins et al., 2009) represented here as a blue dotted line.

**Figure 2**: Beer-Lambert plot used to derive the cross-section of isoprene at 184.95 nm. An unweighted proportional fit to all the data yields $\sigma_{\text{isoprene}}$ = (1.28 $\pm$ 0.02)$\times10^{-17}$ cm$^2$ molecule$^{-1}$, represented by the solid black line on both the main plot and insert. This value (also displayed as the red star on **Figure 1**) was subsequently used to calculate isoprene concentrations for kinetic experiments.

**Figure 3**: Beer-Lambert plots used to derive 184.95 nm absorption cross-sections: $\sigma_{\alpha\text{-pinene}}$ (blue square datapoints) and $\sigma_{\Delta\text{-3-carene}}$ (red circles). Over a wide range of concentrations, good agreement was observed for data from both undiluted samples and from pre-prepared dilute bulbs (section 2). Unweighted proportional fits with (Eq. 1) yield (in units of $10^{-17}$ cm$^2$ molecule$^{-1}$) $\sigma_{\alpha\text{-pinene}}$ = (1.54 $\pm$ 0.05) and $\sigma_{\Delta\text{-3-carene}}$ = (2.40 $\pm$ 0.03).

**Figure 4**: Time-resolved OH LIF signals, $S$, recorded in the presence of different [isoprene] used to determine $k_1$(298 K). Photolysis (R5) of [H$_2$O$_2$] = $2.3\times10^{14}$ molecule cm$^{-3}$ was used to generate [OH] $\approx 5\times10^{11}$ molecule cm$^{-3}$ in $P$ = 103 Torr (N$_2$). Unweighted fits with expression (Eq. 2) yield first-order rate coefficients $B$: for [isoprene] = 0 (black diamonds) $B$ = (513 $\pm$ 12) s$^{-1}$; [isoprene] = $57\times10^{12}$ molecule cm$^{-3}$ (red squares) $B$ = (6654 $\pm$ 44) s$^{-1}$; [isoprene] = $574\times10^{12}$ molecule cm$^{-3}$ (blue circles) $B$ = (53473 $\pm$ 700) s$^{-1}$.

**Figure 5**: Plots of $B$ parameters *vs.* associated [isoprene] used to determine $k_1$ at three different temperatures. Weighted linear fits (Eq. 3) to the data yield (in units of $10^{-11}$ cm$^3$ molecule$^{-1}$ s$^{-1}$): $k_1$(241 K) = (12.9 $\pm$ 0.2) from the blue square datapoints; $k_1$(297 K) = (9.43 $\pm$ 0.14) from the black circles; and $k_1$(356 K) = (6.92 $\pm$ 0.13) from the red triangles. Note that error bars on individual datapoints are too small to be seen here (typically a few hundred on decay rates of several thousand per second).





**Figure 6**: Values of $k_1(T)$ from this work and the literature. A fit of expression (E4) to our data (red squares) yields $k_1$(241-356 K) = $(1.93 \pm 0.08) \times 10^{-11} \exp\{(466 \pm 12)/T\}$ cm$^3$ molecule$^{-1}$ s$^{-1}$, independent of bath-gas pressure and identity (N$_2$ or air). Also displayed are the data from (Campuzano-Jost et al., 2004) presented as published (open blue circles) and subsequently corrected using isoprene absorption cross-sections from (Martins et al., 2009) (solid blue circles, see section 3.2.1 for details). Note that the pressure-dependent dataset from (Park et al., 2004) is here represented by $P \approx 8$ Torr (Ar) results (the highest experimental pressure studied); extrapolations to $k_1$(760 Torr) would yield larger values of $k_1$.

Note that error bars were included for all datasets, but were commonly smaller than the data markers. The dashed black line represents the current IUPAC recommendation of $k_1$ (240 – 430 K) = $2.7 \times 10^{-11} \exp(390/T)$ cm$^3$ molecule$^{-1}$ s$^{-1}$. Not displayed for reasons of clarity are the $T < 220$ K datapoints from (Spangenberg et al., 2004), nor the many literature results at around $T = 298$ K only (see IUPAC).

**Figure 7:** plot of typical LIF profiles recorded for three $\alpha$-pinene concentrations, used to determine $k_2$(296 K). For all experiments photolysis (R4) at $t = 0$ generated [OH] $\approx 5 \times 10^{11}$ molecule cm$^{-3}$ in $P = 97$ Torr (air). Unweighted fits with (E2) yield $B$ parameters: for [$\alpha$-pinene] = $3.9 \times 10^{13}$ molecule cm$^{-3}$ (black squares), $B = (1921 \pm 90)$ s$^{-1}$; [$\alpha$-pinene] = $9.2 \times 10^{13}$ molecule cm$^{-3}$ (red triangles), $B = (4819 \pm 213)$ s$^{-1}$; [$\alpha$-pinene] = $35.6 \times 10^{13}$ molecule cm$^{-3}$ (blue circles), $B = (17944 \pm 615)$ s$^{-1}$.

**Figure 8**: plots of $B$ parameters derived from expression (Eq. 2) *vs.* associated [$\alpha$-pinene] used to determine $k_2$ (OH + $\alpha$-pinene) and $k_3$ (OH + $\Delta$-3-carene) in conditions of $T = 296$ K and $P = 97$ Torr (air). Weighted linear fits yield (in units of $10^{-11}$ cm$^3$ molecule$^{-1}$ s$^{-1}$): $k_2$(296 K) = $(5.3 \pm 0.15)$ from the red squares; $k_3$(296 K) = $(8.1 \pm 0.1)$ (blue circles). Note that 2$\sigma$ statistical errors in $B$, used to weight these fits, were smaller than the symbols in this plot.

**Figure 9**: $k_2(T)$ and $k_3(T)$ from this work in Arrhenius format, together with previous experimental results and a $k_2(T)$ literature evaluation. Not displayed for reasons of clarity are literature datasets containing only ambient temperature $k_2$ results (these are listed in full in **Table 3**), and the 44 determinations of $k_2$(297 – 364 K) from (Gill and Hites, 2002) which are instead represented (purple dotted line) by a fit with (Eq. 4).




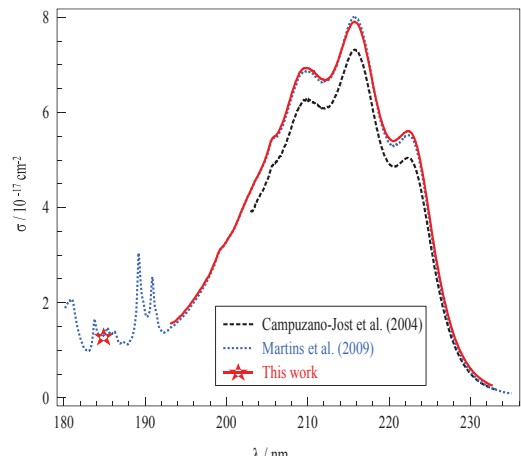

Figure 1





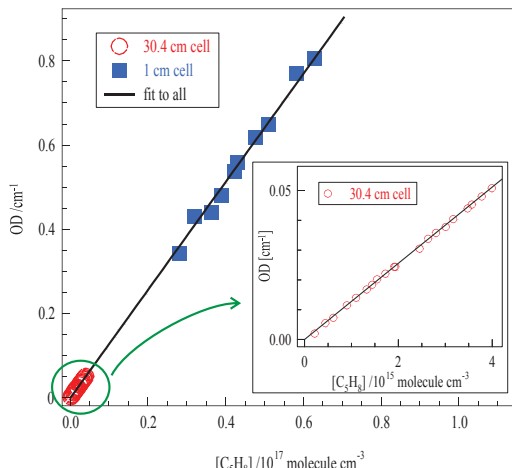

Figure 2





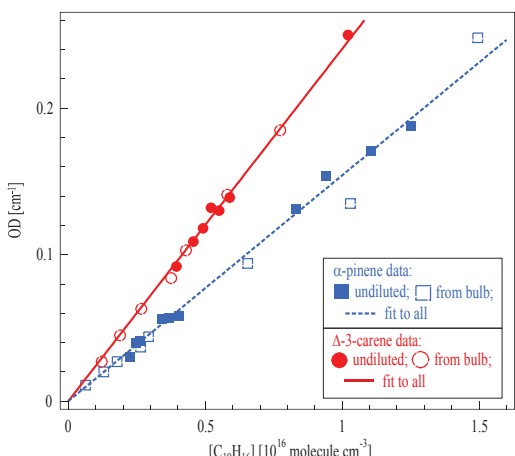

Figure 3





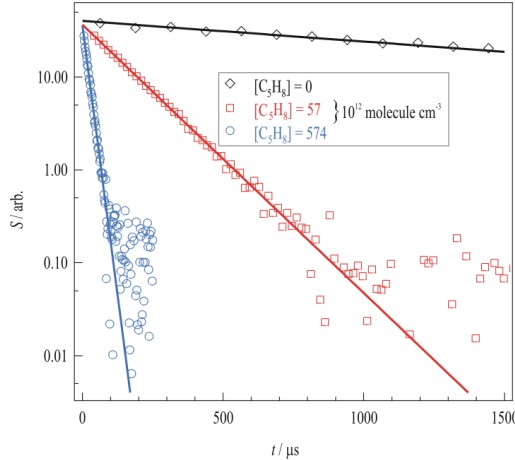

Figure 4



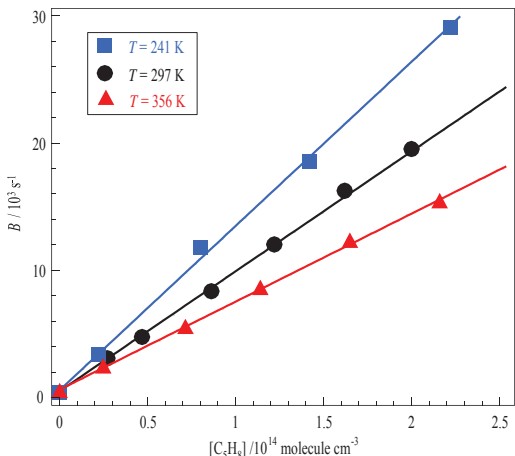

Figure 5





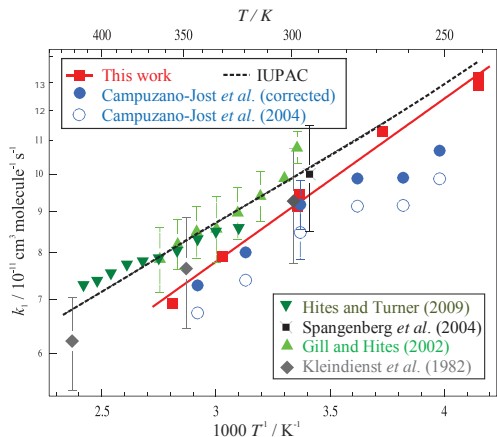

Figure 6





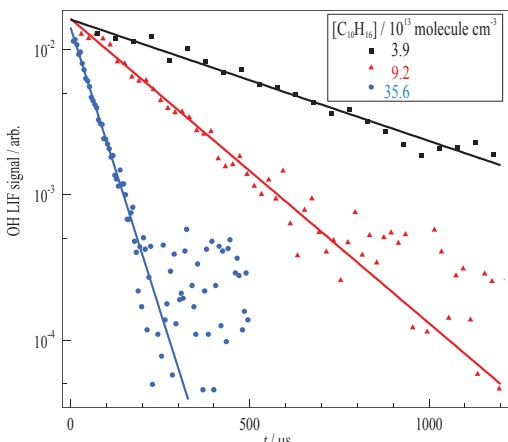

Figure 7




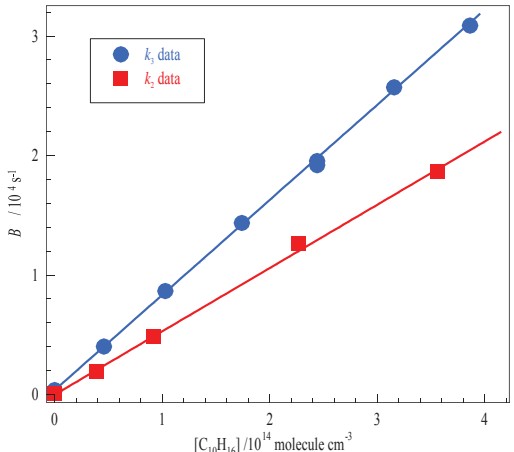

Figure 8





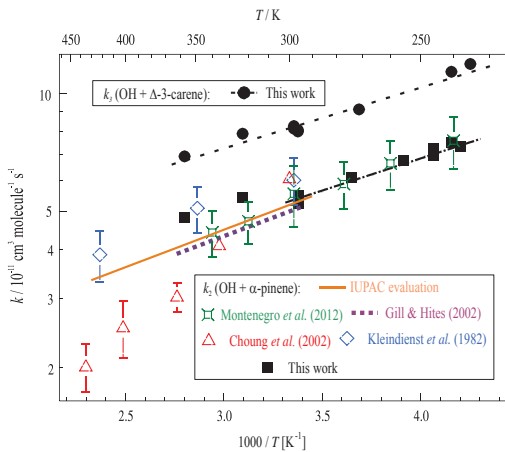

Figure 9