# Peer review of "Temperature dependent rate coefficients for the reactions of the hydroxyl radical with the atmospheric biogenics isoprene, alpha-pinene and delta-3-carene"

_Atmospheric Chemistry and Physics, 2017_

## Referee Comment (RC1) · Anonymous Referee #1 · 16 Jul 2017

**General comments:**

The paper reports the results of a carefully conducted study of three OH reactions important for atmospheric/environmental science along with the analysis of the available data. The subject is relevant to *ACP*. The quality of the obtained data are excellent and the adequate discussion of the experimental procedure, data treatment and results obtained is provided. A manuscript can be accepted for publication in *Atmospheric Chemistry and Physics* after minor revision. Below are comments authors may consider and address in preparation the final version of the manuscript.

**Specific comments:**

Page 4, line 12: Using the "head-space above a liquid sample", i.e. the vapor/gas phase of the sample is potentially a source of an error, especially when impurities in the sample are not identified. The vapor phase can be substantially enriched with more volatile impurities thus affecting results of both spectral and kinetic measurements. The stated purity of reactants (Aldrich) characterizes a bulk liquid sample whereas the amount of impurities in the vapor phase can be substantially different. So, it is always a good idea to use liquid sampling to prepare reactants.

Page 4, lines 13-16: This is confusing a bit. Condensation onto optical windows as well as other surfaces may occur in both cases of using pure compound and diluted mixture. However, when working with pure compound we are aware of the concentration in the gas phase (pressure control), at least. Using absorption cells of different optical path length allows to check for such condensation. When working with diluted mixtures, we do not minimize the potential condensation problem. Moreover, we have no idea on the compound concentration in the gas phase if such condensation occurs. Of course, similar results obtained with two approaches would prove the absence of the problem.

Page 6, line 26: Once small "error" is mentioned, it would be useful to clarify if this is about one st. err., two st. err., 95% confidence interval or anything else.

Page 6 (Eq. 2), page 8 (Eq. 4), Figures 5, 8 and through the text: This is entirely up to authors' discretion because it is simply supporting mathematics. However, why not to use different symbols for different parameters in Eq. 2, Eq. 4, and through the text? At least, "*E/R"* is widely used in the Arrhenius equation (Eq. 4) by both kinetic data makers and users.

Page 7, line 1: Why "scatter" is here? This is probably "change" or "variation" due to variation of laser energy.

Page 7, line 18 and later through the text: Using "non-Arrhenius temperature dependence" to name the temperature dependence, which follows an Arrhenius expression, Eq. 4, with the positive *B* seems very confusing for a gas kinetic paper. The "non-Arrhenius temperature dependence" or non-Arrhenius behavior is rather associated with deviations of the observed data from the Arrhenius dependence given by Eq. 4 in the manuscript. Actually, this "inconsistency" appears in the authors' text. Authors present Eq. 4 as "Arrhenius expression" (page 8, line 26) and use it in deriving the result. In page 11 lines 21-22 authors declare "non-Arrhenius dependence" derived… from the fit of the Arrhenius

expression (Eq. 4) to their data. I would rather suggest using the commonly accepted "negative $E/R$" or authors' "negative $B$".

Page 7, line 30:  How those "two studies of $k_1(298\ K)$ conducted at lower pressures indicated that the high pressure limit was close to 3 Torr and 1 Torr"?

Page 8, line 5:  This is not exactly correct statement. The value (8.47 ± 0.59) overlaps with the range of IUPAC recommendation of $8.7 < k_1(298\ K) < 1.15$. It actually barely overlaps even with the value reported in the present work, (9.3 ± 0.4).
Saying this, it would be beneficial for the discussion of IUPAC recommendation if the IUPAC recommended uncertainty interval is also shown in Figure 5.

Page 8, line 18:  It would be useful to clarify why results of these particular studies are shown in Figure 6. One could guess that only data obtained over the temperature range are shown, but the single room temperature data point reported by Spangenberg et al., 2004 is also shown here.
On the other hand, data from Park et al. are not shown in Figure 6 although they are discussed below (lines 22, 25) and in the figure caption (page 22, lines 6-7).

Page 8, line 25:  IUPAC indicates 249-**348** K as the temperature range of Siese et al., not 249-**438** K. Which one is correct? Also, this paper (from Eurotrac Symposium) is not listed here. Does it make a sense either to list it in the references or not to mention in the discussion?

Page 8, lines 15-17 and 31-33:  Authors suggest re-evaluation of $k_1$ and its uncertainty based on their new data and corrected data from Campuzano-Jost et al., 2004. However, the current IUPAC recommendation is based on results of numerous studies, which yield both the recommended $k_1$ and the uncertainty to overlap the published data chosen for evaluation. I am not going to say that the authors' recommendation makes no sense, but it definitely needs more discussion of the available data, which is probably beyond the scope of this work. So, I would rather suggest a softer statement "probably can be reduced".

Page 10, line 29 and page 11, line 12:  The "black dot-dash line" represents two different data sets – page 25, line 28 and page 11, lines 12-13. Is there any mistake here? In any case, the description of this line (or two lines) is missing in both Figure 9 and the figure caption.

Page 11, line 28:  The statement "…$k(T)$ approaching the collision limit" does not seem very accurate. Authors found that both rate constants increase exponentially with $1/T$; however, there is nothing indicative of approaching to any limiting value yet.

Page 31, Figure 9:  The pink dotted line is described as Gill & Hites in the legend but not shown in Figure 9. Comments on "black dot-dash line" are above (page 10 line 29…)

Page 11, lines 25-34:  Finally, the following is rather discussion/speculations than a comment…
Whereas one can expect both reactions to be similarly fast to a zero approximation, their rate constants are not supposed to be equal.
To a first approximation, one can consider both compounds, $\alpha$-pinene and $\Delta$-3-carene, as derivatives from 1-methylcyclohexene with some possible "steric restrictions" shading $\pi$-electrons of the reactive C=C site. (See their molecular structures, for example, in webbook.nist.com). Moreover, the C=C site is "hidden" a bit better in case of alpha-pinene with the $(CH_3)_2C\langle\rangle CH_2$ cluster located out of plane and closer to the reactive C=C.

Based on these speculations, one could expect the rate constant decreasing from 1-methylcyclohexene to $\Delta$-3-carene ($k_3$) and to $\alpha$-pinene ($k_2$). This is consistent with now available data: $9.8 \times 10^{-11}$ from Aschmann et al., 2012 and $k_3 = 8.1 \times 10^{-11}$, $k_2 = 5.4 \times 10^{-11}$ from the present study. Thus, the values of $k_2$ and $k_3$ determined in the present study seem very reasonable or, better to say, they support the above speculations.

**Technical corrections:**

Title:  It seems like semi-column is missing after "biogenics" – it should be either "…the reactions of the hydroxyl radical with the atmospheric biogenics**:** isoprene, alpha-pinene…" or "…the reactions of the hydroxyl radical with isoprene, alpha-pinene …".

Page 2, line 14:  It seems like something is missing here. I can guess that it was supposed to be as "… Stone et al., 2011) **with** the greatest discrepancies…"

Page 7, line 18:  Figure **6**, not Figure 5.

Page 8, line 26:  Figure **6**, not Figure 8.

Page 9, line 28:  Table **3**, not Table 2.

Page 9, line 31:  "using an absorption cell with the path length of 43.8 cm" would be more appropriate.

Page 10, line 3:  Table **2**, not Table 1.

Page 10, line 7:  Table **3**, not Table 2.

---

## Referee Comment (RC2) · A. Hynes (Referee) · 6 Aug 2017

ACP Title: Temperature dependent rate coefficients for the reactions of the hydroxyl radical with the atmospheric biogenics isoprene, ïĄạ-pinene and ïĄĎ-3-carene Author(s): Terry J. Dillon et al. MS No.: acp-2017-561

Reviewer: A. J. Hynes, University of Miami

I am submitting this review without reading other reviews/comments and I apologize for any duplication. This is a high quality determination of the rate coefficients for the reactions of OH with isoprene, alpha-pinene and delta-3-carene. Temperature, pressure

and bath gas dependencies are measured. The work certainly merits publication but I have a couple of comments that relate to the isoprene study. The difference between the value reported here and two prior "low values" reported by my research group are small with current measurement of 9.3 ± 0.4 overlapping our 2004 value of 8.5 ± 0.6 within the stated uncertainties. I absolutely concur (perhaps self-servingly) with the conclusion that the current IUPAC recommendation is too high. Because of the importance of this reaction in the atmosphere the accuracy and precision of determinations is very important and in the case the difference between the Mainz and Miami work is clearly associated with the value of the absorption coefficient used for in-situ monitoring of reactant concentration. I believe than in-situ monitoring improves the precision of rate coefficient determinations and, as noted in the paper, a subsequent resolution of the discrepancies in the absorption coefficients allows the rate coefficient to be updated. The abstract contains the statement: "These results indicate that significant discrepancies between previous absolute and relative rate determinations of k1 result in part from $\sigma$ values used to derive the isoprene concentration." However it seems to me that the difference in $\sigma$ values only explains the difference between the direct studies from Miami and Mainz, I don't see it having any bearing on the other direct and relative rate studies.

It should be noted that in our work we measured cross sections and rate coefficients for both normal and deuterated isoprene and the spectra were almost identical but with a slight blue shift in the deuterated spectrum. This, however, produced significantly different cross sections at the 228.8 nm Cd atomic line that was used to monitor concentrations in our work. The cross section we measured at 215.5 was within 4% on an old study by Jones and Taylor. I suspect than in this case it may be easier to measure the rate coefficient than the absorption cross-section. We obtained the same rate coefficient (within experimental error) for both compounds and felt this was consistent with an addition to a double bond i.e. there should not be a kinetic isotope effect. I should also note that both of the papers: Campuzano-Jost, P., Williams, M. B., D'Ottone, L. and Hynes, A. J.: Geophys. Res. Lett., 27, 693, 2000. Campuzano-Jost, P., Williams,

M. B., D'Ottone, L. and Hynes, A. J.: J. Phys. Chem. A, 108, 1537, 2004. contain several confusing errors about the actual wavelength that was used to monitor concentration. As the corresponding author this was my responsibility and I apologize to others, including the current authors, who had to decipher this. It was, as noted above, the 228.8 nm Cd atomic line.

For reactions (2) and (3) that reported values confirm and extend the current literature database. Again, this is a high quality study that merits publication.

Please also note the supplement to this comment:
https://www.atmos-chem-phys-discuss.net/acp-2017-561/acp-2017-561-RC2-supplement.pdf

---

## Author Comment (AC1) · 2 Oct 2017

**Final author comments on manuscript acp-2017-561**

**"Temperature dependent rate coefficients for the reactions of the hydroxyl radical with the atmospheric biogenics isoprene, $\alpha$-pinene and $\Delta$-3-carene"**

All authors wish to express their appreciation to Dr. Tony Hynes and the anonymous reviewer for their careful consideration of the manuscript. There follows a list of reviewer comments (in black), together with our response and details of corrections / improvements to the manuscript (in blue).

**From the anonymous referee in acp-2017-561-RC1**

*"Page 4, line 12: Using the "head-space above a liquid sample", i.e. the vapor/gas phase of the sample is potentially a source of an error, especially when impurities in the sample are not identified. The vapor phase can be substantially enriched with more volatile impurities thus affecting results of both spectral and kinetic measurements. The stated purity of reactants (Aldrich) characterizes a bulk liquid sample whereas the amount of impurities in the vapor phase can be substantially different. So, it is always a good idea to use liquid sampling to prepare reactants."*

The reviewer is correct that the chemical as supplied may contain volatile impurities that in the vapour phase amount to a substantial proportion of the head-space. However, as stated at the end of the experimental section, all organics were purified by repeated freeze-pump-thaw cycles at $T$ = 77 K, which would have depleted the proportion of more volatile impurities. It is also worth noting that the compound for which the cross-section values are most at issue is isoprene, which itself has a large vapour pressure of 8.8 psi at 293 K (Aldrich). Isoprene is consequently not likely to be significantly impacted by other more volatile organics. A possible improvement for any future work would be for BOTH head-space and liquid-sampling methods to be used when preparing samples.

*"Page 4, lines 13-16: This is confusing a bit. Condensation onto optical windows as well as other surfaces may occur in both cases of using pure compound and diluted mixture. However, when working with pure compound we are aware of the concentration in the gas phase (pressure control), at least. Using absorption cells of different optical path length allows to check for such condensation. When working with diluted mixtures, we do not minimize the potential condensation problem. Moreover, we have no idea on the compound concentration in the gas phase if such condensation occurs. Of course, similar results obtained with two approaches would prove the absence of the problem."*

It is possible that condensation could occur from both undiluted and dilute samples. The limiting factor in these experiments was the measurement of small gas phase pressures. When using undiluted samples, a large concentration of VOC was required to attain a measurable pressure (see the blue square datapoints on Figure 2). A small 1 cm absorption cell was consequently necessary to limit the extent of absorption from these high concentrations. This combination of large concentrations and a cell with a high surface area to volume ratio is potentially prone to large errors from condensation. By contrast, small concentrations and a larger (30 cm) cell with a consequently smaller surface area to volume ratio were able to be used in conjunction with dilute VOC samples. As the reviewer points out *"… similar results obtained with two approaches would prove the absence of the*

*problem."* Such good agreement between the two methods is what is presented in Figures 2 & 3 of the paper, and was commented upon in section 3.1. Nonetheless, to avoid confusion we have amended the text on page 3 after line 13 to read

*"These two supply methods had complementary advantages and disadvantages. Use of undiluted samples was more direct, but with uncertainties due to the measurement of small sample pressures, and the potential for VOC (present at high concentrations) to condense onto the windows of the 1 cm absorption cell (high surface area to volume ratio). These problems were minimized when using dilute VOC at higher pressure in the 30 cm cell (with relatively small surface area to volume ratio) but with the potential for errors associated with sample preparation and storage."*

*"Page 6, line 26: Once small "error" is mentioned, it would be useful to clarify if this is about one st. err., two st. err., 95% confidence interval or anything else."*
The opening paragraph of section 3 states that all errors quoted are two standard error values. Nonetheless, we have reiterated this in the text on page 6 line 26 to read *"(two standard errors were generally less than 2%)."*

*"Page 6 (Eq. 2), page 8 (Eq. 4), Figures 5, 8 and through the text: This is entirely up to authors' discretion because it is simply supporting mathematics. However, why not to use different symbols for different parameters in Eq. 2, Eq. 4, and through the text? At least, "E/R" is widely used in the Arrhenius equation (Eq. 4) by both kinetic data makers and users."*
We agree, and have replaced B in the Arrhenius expression (eq4) and throughout the text with the commonly used $\frac{E}{R}$

*"Page 7, line 1: Why "scatter" is here? This is probably "change" or "variation" due to variation of laser energy."*
We have replaced "scatter" with "variation"

*"Page 7, line 18 and later through the text: Using "non-Arrhenius temperature dependence" to name the temperature dependence, which follows an Arrhenius expression, Eq. 4, with the positive B seems very confusing for a gas kinetic paper. The "non-Arrhenius temperature dependence" or non-Arrhenius behavior is rather associated with deviations of the observed data from the Arrhenius dependence given by Eq. 4 in the manuscript. Actually, this "inconsistency" appears in the authors' text. Authors present Eq. 4 as "Arrhenius expression" (page 8, line 26) and use it in deriving the result. In page 11 lines 21-22 authors declare "non-Arrhenius dependence" derived... from the fit of the Arrhenius expression (Eq. 4) to their data. I would rather suggest using the commonly accepted "negative E/R" or authors' "negative B"."*
The term "non-Arrhenius temperature dependence" is commonly used in kinetics publications to denote a rate coefficient that does not follow the model of Arrhenius. In such cases, the path from reactants to products is not hindered by a large energetic barrier; any changes $k(T)$ observed may, for example, be caused by intermediate complex formation along the reaction coordinate or simply changes in collision frequency. The rate coefficients in this paper therefore all exhibit a non-Arrhenius temperature dependence. It is however conventional to plot such data in Arrhenius format, hence the conventions followed in this manuscript. For any readers not familiar with the conventions of kinetics, the meaning of

the phrase "non-Arrhenius" was explained on page 7 line 19 as "The data displayed in **Figure 5** further reveal the "negative" or non-Arrhenius temperature dependence for $k_1$, with the largest slopes obtained in experiments conducted at the lowest temperatures (241 K)."

*"Page 7, line 30: How those "two studies of k1(298 K) conducted at lower pressures indicated that the high pressure limit was close to 3 Torr and 1 Torr"?*
This line has been amended to read *"Results from two studies of $k_1$(298 K) conducted at lower pressures indicate that the high-pressure limit was close to 3 Torr (He) (Singh and Li, 2007) and 1 Torr (He) (Chuong and Stevens, 2000)."*

*Page 8, line 5: This is not exactly correct statement. The value (8.47 ± 0.59) overlaps with the range of IUPAC recommendation of 8.7 < k1(298 K) < 1.15. It actually barely overlaps even with the value reported in the present work, (9.3 ± 0.4). Saying this, it would be beneficial for the discussion of IUPAC recommendation if the IUPAC recommended uncertainty interval is also shown in Figure 5.*
Many thanks for spotting this. We have amended the text on page 8 to read *"…lower than most other determinations, and at the lower limit of the range quoted in the IUPAC recommendation."*

*Page 8, line 18: It would be useful to clarify why results of these particular studies are shown in Figure 6. One could guess that only data obtained over the temperature range are shown, but the single room temperature data point reported by Spangenberg et al., 2004 is also shown here. On the other hand, data from Park et al. are not shown in Figure 6 although they are discussed below (lines 22, 25) and in the figure caption (page 22, lines 6-7).*
The dataset from Park, and those from (Singh and Li, 2007) and (Chuong and Stevens, 2000) describe rate coefficients in a low pressure regime that consequently change with pressure (and temperature). There is no simple way to present these on Figure 6, which focusses on data in a high-pressure limiting regime where the values of $k(T)$ are useful for atmospheric chemistry. We have modified the caption to Figure 6 to now read *"Not displayed for reasons of clarity are the T < 220 K datapoints from (Spangenberg et al., 2004), results from low-pressure studies of (Park et al., 2004) nor the many literature results at around T = 298 K only (see IUPAC)."*

*Page 8, line 25: IUPAC indicates 249-348 K as the temperature range of Siese et al., not 249-438 K. Which one is correct? Also, this paper (from Eurotrac Symposium) is not listed here. Does it make a sense either to list it in the references or not to mention in the discussion?*
The dataset from Siese does indeed extend to 438 K, as stated in this manuscript. We will include the Eurotrac citation in the references here.

*"Page 8, lines 15-17 and 31-33: Authors suggest re-evaluation of k1 and its uncertainty based on their new data and corrected data from Campuzano-Jost et al., 2004. However, the current IUPAC recommendation is based on results of numerous studies, which yield both the recommended k1 and the uncertainty to overlap the published data chosen for evaluation. I am not going to say that the authors' recommendation makes no sense, but it definitely needs more discussion of the available data, which is probably beyond the scope of this work. So, I would rather suggest a softer statement "probably can be reduced".*

Agreed, we have therefore amended the text on page 8 to read "could indicate that this evaluation is quoted with an overly conservative uncertainty." and "recommended uncertainties could be reduced in any subsequent re-evaluation by IUPAC."

*Page 10, line 29 and page 11, line 12: The "black dot-dash line" represents two different data sets – page 25, line 28 and page 11, lines 12-13. Is there any mistake here? In any case, the description of this line (or two lines) is missing in both Figure 9 and the figure caption.*
The black dot dashed line was intended to represent only our data on Figure 9. The figure, legend and caption have been amended to this effect. Reference to this line on page 11 in the context of the Montenegro data was indeed confusing. The issue has been clarified by the replacement of text on page 11 lines 12-13 with *"It is clear from **Figure 9** that these two datasets could provide a basis for IUPAC to extent the recommendation for $k_2$ to T < 290 K."*

*Page 11, line 28: The statement "…k(T) approaching the collision limit" does not seem very accurate. Authors found that both rate constants increase exponentially with 1/T; however, there is nothing indicative of approaching to any limiting value yet.*
The rate coefficients measured here are all close to $10^{-10}$ $cm^3$ $molecule^{-1}$ $s^{-1}$, and therefore approaching collision-limiting values. For clarification, the text on page 11 has been amended to read *"… values approaching the collision limit of $10^{-10}$ $cm^3$ $molecule^{-1}$ $s^{-1}$."*

*Page 31, Figure 9: The pink dotted line is described as Gill & Hites in the legend but not shown in Figure 9. Comments on "black dot-dash line" are above (page 10 line 29…)*
The line is displayed on Fig 9 as published in ACPD. The colour was perhaps better described as purple, and we have chosen a more distinctive deep purple in the new version of Fig 9.

*Page 11, lines 25-34: Finally, the following is rather discussion/speculations than a comment…*
*Whereas one can expect both reactions to be similarly fast to a zero approximation, their rate constants are not supposed to be equal. To a first approximation, one can consider both compounds, α-pinene and Δ-3-carene, as derivatives from 1-methylcyclohexene with some possible "steric restrictions" shading p-electrons of the reactive C=C site. (See their molecular structures, for example, in webbook.nist.com). Moreover, the C=C site is "hidden" a bit better in case of alpha-pinene with the (CH3)2C<>CH2 cluster located out of plane and closer to the reactive C=C. Based on these speculations, one could expect the rate constant decreasing from 1-methylcyclohexene to Δ-3-carene (k3) and to a-pinene (k2). This is consistent with now available data: $9.8 \times 10^{-11}$ from Aschmann et al., 2012 and k3=$8.1 \times 10^{-11}$, k2=$5.4 \times 10^{-11}$ from the present study. Thus, the values of k2 and k3 determined in the present study seem very reasonable or, better to say, they support the above speculations.*
A very insightful comment, and one we will surely see followed up in future SAR development.

**Technical corrections:**
*Title: It seems like semi-column is missing after "biogenics" – it should be either "…the reactions of the hydroxyl radical with the atmospheric biogenics: isoprene, alpha-pinene…" or "…the reactions of the hydroxyl radical with isoprene, alpha-pinene …".*

We did consider "… the atmospheric biogenics: isoprene; alpha-pinene; …" using a colon and semi-colons, but decided this was getting too much punctuation for a title. After all, we don't include a full stop either. Happy to take advice from the editorial team on this.

*Page 2, line 14: It seems like something is missing here. I can guess that it was supposed to be as "… Stone et al., 2011) with the greatest discrepancies…"*
Many thanks for spotting this. Now *"… Stone et al., 2011), with the greatest…"*

*Page 7, line 18: Figure 6, not Figure 5.*
No, this is meant to refer to Figure 5 (though the same trend is also evident from Figure 6)

*Page 8, line 26: Figure 6, not Figure 8.*
Many thanks, correction made

*Page 9, line 28: Table 3, not Table 2.*
No, this is meant to refer to Table 2. To clarify matters, we have amended the text to read *"Results from these previous studies, together with the $k_2$ values from this work (themselves listed in **Table 2**) are summarised in **Table 3** and depicted on **Figure 9**."*

*Page 9, line 31: "using an absorption cell with the path length of 43.8 cm" would be more appropriate.*
This passage refers to values of parameters used in (Eq 1), rather than the apparatus. To clarify we have amended the text to read *"Optical signals were converted (E1) to absolute [Δ-3-carene] using values of l = 43.8 cm and $\sigma_{\Delta\text{-3-carene}} = 2.4\times10^{-17}$ cm$^2$ molecule$^{-1}$ as determined in this work."*

*Page 10, line 3: Table 2, not Table 1.*
Many thanks for spotting this – correction made.

*Page 10, line 7: Table 3, not Table 2.*
Many thanks for spotting this – correction made.

**From Dr. Hynes in acp-2017-561-RC2**

*This is a high quality determination of the rate coefficients for the reactions of OH with isoprene, alpha-pinene and delta-3-carene. Temperature, pressure and bath gas dependencies are measured. The work certainly merits publication but I have a couple of comments that relate to the isoprene study. The difference between the value reported here and two prior "low values" reported by my research group are small with current measurement of 9.3 ± 0.4 overlapping our 2004 value of 8.5 ± 0.6 within the stated uncertainties. I absolutely concur (perhaps self-servingly) with the conclusion that the current IUPAC recommendation is too high. Because of the importance of this reaction in the atmosphere the accuracy and precision of determinations is very important and in the case the difference between the Mainz and Miami work is clearly associated with the value of the absorption coefficient used for in-situ monitoring of reactant concentration. I believe than in-situ monitoring improves the precision of rate coefficient determinations and, as noted in the*

*paper, a subsequent resolution of the discrepancies in the absorption coefficients allows the rate coefficient to be updated. The abstract contains the statement: "These results indicate that significant discrepancies between previous absolute and relative rate determinations of k1 result in part from values used to derive the isoprene concentration." However it seems to me that the difference in values only explains the difference between the direct studies from Miami and Mainz, I don't see it having any bearing on the other direct and relative rate studies.*

The statement in the abstract is that these differences in the literature were in part due to sigma values used to derive the isoprene concentration. This is a key finding of this work, where systematically larger cross-sections for isoprene were determined than those from Dr. Hynes' publications (see main manuscript Fig. 1). We shall clarify this point by adding to the text so it now reads

*"These results indicate that significant discrepancies between previous absolute and relative rate determinations of $k_1$ result in part from $\sigma$ values used to derive the isoprene concentration in high-precision absolute determinations."*

There may well be other factors at play, such as the values used to place relative rate data on an absolute scale, but these factors were not investigated in this work.

*It should be noted that in our work we measured cross sections and rate coefficients for both normal and deuterated isoprene and the spectra were almost identical but with a slight blue shift in the deuterated spectrum. This, however, produced significantly different cross sections at the 228.8 nm Cd atomic line that was used to monitor concentrations in our work. The cross section we measured at 215.5 was within 4% on an old study by Jones and Taylor. I suspect than in this case it may be easier to measure the rate coefficient than the absorption cross-section. We obtained the same rate coefficient (within experimental error) for both compounds and felt this was consistent with an addition to a double bond i.e. there should not be a kinetic isotope effect.*

The lack of kinetic isotope effect is certainly consistent with the reaction mechanisms discussed in this work. However, it is possible and indeed likely that any small systematic errors in the determination of an isoprene cross section would be present in similar studies of deuterated isoprene.

*I should also note that both of the papers: Campuzano-Jost, P., Williams, M. B., D'Ottone, L. and Hynes, A. J.: Geophys. Res. Lett., 27, 693, 2000. Campuzano-Jost, P., Williams, M. B., D'Ottone, L. and Hynes, A. J.: J. Phys. Chem. A, 108, 1537, 2004. contain several confusing errors about the actual wavelength that was used to monitor concentration. As the corresponding author this was my responsibility and I apologize to others, including the current authors, who had to decipher this. It was, as noted above, the 228.8 nm Cd atomic line. For reactions (2) and (3) that reported values confirm and extend the current literature database. Again, this is a high quality study that merits publication.*

Many thanks to Dr. Hynes and Reviewer 1 for their supportive comments.

---

## Author Response (AR1)

Dear ACP editorial team

please find here an upload of our revised manuscript and abstract. Changes were detailed in the author's response file that was dealt with previously.

Please also note that we need the online title for this paper changing such that symbols are removed and replaced with full words "alpha" and "delta". Further, one of our authors is listed incorrectly as "Christoph MB Gross" when he should be "Christoph BM Gross".

Many thanks if you can help with these changes. Best Wishes, Terry